# Responses of Soil Microbial Communities in Soybean–Maize Rotation to Different Fertilization Treatments

Yunlong Wang [1], Liqiang Zhang [1], Fangang Meng [2], Zixi Lou [1], Xiaoya An [1], Xinbo Jiang [1], Hongyan Zhao [1,*] and Wei Zhang [2,*]

1 College of Agronomy, Yanbian University, Yanji 133002, China; 2021050850@ybu.edu.cn (Y.W.); 2020010597@ybu.edu.cn (L.Z.); 2021050830@ybu.edu.cn (Z.L.); 2021010832@ybu.edu.cn (X.A.); 2021010595@ybu.edu.cn (X.J.)
2 Soybean Research Institute, Jilin Academy of Agricultural Sciences, National Engineering Research Center of Soybean, Changchun 130033, China; mengfg2013@163.com
* Correspondence: zhy@ybu.edu.cn (H.Z.); zw.0431@163.com (W.Z.)

**Abstract:** Rotation and fertilization are important methods used to improve crop yield. In particular, crop rotation is an effective means of enhancing ecosystem diversity; however, there exist relatively few studies regarding the effects of long-term maize–soybean rotation and fertilization on soil microbial communities. To further understand the changes in soil microbial community structure under long-term maize–soybean rotation and fertilization, we used a 9-year-old experimental site with maize–soybean rotation as the research object and soybean continuous cropping as a control. We explored the growth effects of soybean and the changes in soil microbial communities under the soybean–maize rotation system and fertilization treatments by analyzing the physicochemical properties of the soil, crop agronomic traits, yield, and changes in soil microbial community structure. The results show that, in comparison with soybean continuous cropping, the yield of soybeans was increased by 12.11% and 21.42% under maize–soybean rotation with different fertilization treatments, respectively. Additionally, there was a significant increase in the agronomic effects of nitrogen following rotation combined with fertilization. Moreover, the soil pH, SOM, and nutrient status were also improved. Bryobacter, Gemmatimonas, and Rhodanobacter were the dominant bacteria. Rotation treatment increased the relative abundance of Bryobacter and Rhodanobacter, and fertilization treatment increased the relative abundance of Gemmatimonas. Rotation also increased the stability of the bacterial community structure and strengthened the symbiotic relationship between species. The prediction of nitrogen-related functional genes indicates that rotation increased soil ammonification and nitrification. Heterocephalacria and Mrakia were the dominant fungal genera under crop rotation. The abundance of Saccharomyces Mrakia was significantly positively correlated with ammonium nitrogen levels and crop yield. Crop rotation increased the abundance of Saccharomyces Mrakia and reduced the abundance of Fusarium, but fertilization increased the abundance of Fusarium. Functional gene prediction also indicates that the relative abundance of plant pathogens was significantly reduced. This study provides a theoretical basis for soil microbial diversity and ecosystem service function in long-term soybean–maize rotation.

**Keywords:** soybean–maize rotation; fertilization; soil microbial communities

## 1. Introduction

The planting pattern of intensively used farmland is one of the most important factors affecting soil's physical, chemical, and biological properties. Changes in the planting pattern will inevitably lead to changes in soil fertility and nutrients [1]. Rotation is a key method to alleviate the continuous cropping barrier, rationally utilize soil nutrients, coordinate crop nutrient absorption, and balance soil nutrients [2]. Understanding the effects of rotation on crops is therefore critical for the development of nutrient management

strategies to optimize yields, while maintaining the sustainability of cropping patterns. The rotation of leguminous and gramineous crops is the essence of traditional agriculture, which can effectively overcome the obstacles of soybean continuous cropping [3] and drastically reduce the occurrence of disease, insect pests, and weeds in soybean production. Rotation maintains soil health, increases soil microbial diversity, and improves soil microbial activity. In particular, proper rotation of crops can reduce the rate of chemical fertilizer use [4,5] and improve the utilization efficiency of nitrogen and other nutrients to promote yield [6,7]. However, the effects of different fertilization rates on soil properties and crop yield in soybean rotation, especially the contribution of soil microorganisms, have not been well studied.

The importance of soil microbial communities in the regulation of farmland structure and function has received increasing attention from the ecological field [8]. Research has confirmed that the effects of different fertilization types and rates on the composition of soil bacterial and fungal communities are different [9,10]. The response of microbial communities to long-term nutrient fertilization greatly regulates plant production in agroecosystems [11,12]. The relationship between microbial community structure and diversity under different fertilization methods has been studied. Based on a 35-year long-term fertilization experiment, it was confirmed that fertilization can directly affect soil nitrogen fixation activity, but also indirectly by influencing soil physicochemical properties and nitrogen-fixing bacterial communities [13–15]. Many studies have indicated the effects of different fertilization methods on crop yield and soil physicochemical properties; however, there remains a lack of understanding of the mechanism underlying the relationship among soil microbial communities, crops, and nutrient fertilization. Therefore, studying the regulation of soil microbial community structure, following different fertilization methods in relation to soil physicochemical properties, crop yield, and agronomic traits, is meaningful for the reduction in fertilization rates and improvement in soil ecological environments.

Soil microorganisms can regulate soil microecology, promote the circulation of material and nutrient elements of minerals, and promote the diversity of soil nutrients [16–19]. Soil microorganisms are closely related to soil physicochemical properties and crop growth, and participate in the decomposition of soil organic matter, the formation of soil humus, soil material transformation, and nutrient cycling. At the same time, the composition of soil microbial communities and changes in soil microbial abundance can be used to evaluate soil quality, fertility, and crop productivity. Many studies have shown that the cropping system is the main factor affecting microbial community structure [10,15,20], which is also changed by crop rotation [21]. One study found that it was only the abundance of Actinobacteria in soybean–maize rotation that was significantly higher than that in maize rotation [22], and another study reported that the abundance of Actinomycetes and Firmicutes in the wheat–maize–soybean rotation system increased significantly in comparison with that in soybean continuous cropping [6]. Other studies have highlighted that changes in soil community structure are related to changes in soil total nitrogen, organic matter, available nitrogen, phosphorus, and soil pH [23]. An evaluation of the parameters of soybean–maize rotation and soybean monoculture demonstrated that crop rotation is one of the most promising cultivation methods to improve soil microbial diversity and nutrient cycling, which is beneficial to the growth of crops [24]. Moreover, it is expected that the soil microbial abundance in continuous maize cropping will be decreased in comparison with that in soybean–maize rotation. It has been demonstrated that different soybean planting patterns can affect soil microbial function and life history strategies by changing soil nutrients [25]; therefore, it is necessary to explore the changes in soil microbial communities under a soybean rotation system.

Crop rotation is an ancient farming system with a significant amount of evidence indicating its ability to improve crop yield. However, the mechanisms underlying improvements in crop yield and soil physicochemical properties by long-term crop rotation, and the manner by which changes in microbial community occur, are less studied. Therefore, the present study evaluates the 10-year soybean–maize rotation cropping system following

different fertilization and non-fertilization methods with a view to exploring the ecological relationship among soil properties, soybean yield, agronomic traits, and microbial communities and functions. We reveal the succession of soil microbial communities under the long-term rotation of soybean at different growth stages, furthering our understanding of the ecological functions of soil microorganisms.

## 2. Materials and Methods

### 2.1. Experimental Site and Design

This experiment was conducted in April 2019 at the Yanming Lake Seed Company base (128°21′33″ E; 43°26′24″ N) located in Guandi Town, Dunhua city, Yanbian Korean Autonomous Prefecture, Jilin Province. The soil type is medium level albic, and the soil fertility grade is three. Soybean (Glycine max, 'Deyu 576') and maize (Zea mays, 'Jiyu 47') were used in this experiment. The base fertilizer provided for maize is N, $P_2O_5$, and $K_2O$ at 75 kg/hm$^2$, 90 kg/hm$^2$, and 75 kg/hm$^2$, respectively. The topdressing for maize was provided at the V12 stage. The base fertilizer provided for soybean is N, $P_2O_5$, and $K_2O$ at 60 kg/hm$^2$, 75 kg/hm$^2$, and 75 kg/hm$^2$, respectively.

The field experiment included four treatments: (1) maize–soybean rotation with regular fertilization (RC1); (2) maize–soybean rotation without fertilization (RC0); (3) soybean continuous cropping with fertilization (CC1); and (4) soybean continuous cropping without fertilization (CC0). All treatments were performed in triplicate. The planting area of each group was 36 m$^2$; 12 rows of crops were planted in each group; the length of each row was 5 m; and the row spacing was 0.6 m.

### 2.2. Soil Sample Collection

Soil was randomly collected from around 10 soybean plants for each treatment (in triplicate) and mixed as one sample. The collected soil samples were placed in an ice box and transported to the laboratory, where plant roots, plant residues, and stones were removed. After sieving through a 2 mm mesh, one part of the rhizosphere soil samples was stored at −80 °C until total DNA extraction and the other part was dried at room temperature for the determination of soil physicochemical properties.

### 2.3. Measurement of Soil Properties

Soil pH and EC (electrical conductivity) were measured in deionized water at a soil:water ratio of 1:5 using a pH meter (SX-620, Leizi, Shanghai, China) and an EC meter (DDSJ-308A, Leizi, Shanghai, China), respectively. The soil organic carbon (SOC) was subjected to additional heat treatment following the addition of $Cr_2O_7$ solution to react with the carbon in the soil sample and titrated with $FeSO_4$ for 24 h. The Fang method was used to measure AP and TP [26], while AK and TK were measured using flame emission spectrometry. Air-dried soil samples were extracted with 1 M $CaCl_2$ to determine AN and NN using a previously described protocol [27], while TN was measured using the Kjeldahl method.

### 2.4. Agronomic Indexes and Yield

After crop harvest, the agronomic indexes, such as plant height, stem diameter, and stem node number, were determined. Meanwhile, 10 representative plants were selected from each experimental plot to determine the total grain weight per plant and 100-grain weight of soybean, and the theoretical yield was calculated. Agronomic efficiency of nitrogen fertilizer is expressed as the ratio of crop yield to the amount of nitrogen fertilizer applied.

### 2.5. Microbiological Analysis

Genomic DNA was extracted from each soil sample using a Fast E.Z.N.A.® Kit for Soil (Omega Bio-tek, Norcross, GA, USA) according to the manufacturer's instructions. The abundance of total bacterial and fungal communities was quantitated by qPCR of 16S rDNA

and 18S rDNA, respectively. DNA was purified using an AxyPrep DNA Gel Extraction Kit (Axygen Biosciences, Union City, CA, USA). The universal primer gene sequences used were 338F (5′-ACTCCTACGGGAGGCAGCA-3′), 806R (5′-GGACTACHVGGGTWTCTAAT-3′), 357F (5′-CCTACGGGAGGCAGCAG-3′), and 518R (5′-ATTACCGCGGCTGCTGG-3′). Three replicates were carried out for each sample. Sequencing was performed using the Miseq PE300 platform of Illumina Inc. (San Diego, CA, USA) [27]. Table 1 shows the soil microbial analytical methods.

**Table 1.** Soil microbial analytical methods.

| Analytical Software/Database | Version Number | Use |
| --- | --- | --- |
| Uparse | 7.0.1090 | OTU clustering |
| RDP Classifier | 2.11 | Sequence classification annotation |
| Usearch | 7 | OTU statistics |
| Mothur | 1.30.2 | Analysis of alpha diversity |
| PICRUSt | 1.1.0 | KEGG, COG, and Pfam functional predictions of the 16S sequence |
| SILVA | 138 | rRNA database |
| UNITE | 8 | Fungal ITS database |
| FunGene | 9.6 | Functional gene database |
| MaarjAM | 81 | Fungal 18S rRNA database |
| HPB | -- | 16S rRNA database of human pathogens |
| Funguild | 1 | Database of fungal functional annotation |
| MAFFT | 7.2 | Multiple sequence alignment |

### 2.6. Data Analysis

SPSS 20.0 was used for statistical analysis. The collation of experimental data and creation of graphs were performed using GraphPad Prism 9. Differences in soil microbial diversity indexes between treatments were compared using the Meiji platform. The igraph and vegan toolsets in the R 4.2.1 environment were used to organize and plot data.

## 3. Results

### 3.1. Effect of Soybean–Maize Rotation and Fertilization Treatments on Soil Properties

Rotation and fertilization have certain effects on soil physicochemical indicators [28]. As can be seen from Table 2, the EC value of CC1 soil was the highest, followed by RC1, RC0, and CC0 soil. Moreover, the EC value of CC1 soil was 1.6 times higher than that of RC1 soil, indicating that rotation effectively reduced the soil salt ion concentration. The pH of RC0 and CC0 soil was higher than that of RC1 and CC1 soil, and the pH of RC0 soil was almost 0.5 times higher than that of CC1 soil, indicating that rotation improved soil pH, which is in accordance with previous studies. The organic matter content of RC1 and RC0 soil was higher than that of CC1 and CC0 soil, suggesting that rotation was more conducive to the accumulation of soil nutrients by soybean. The total phosphorus and available phosphorus content of RC0 soil was higher than that of RC1 soil. Fertilization after rotation promoted the absorption of phosphorus by plants. The total nitrogen and total potassium content of RC0 soil was higher than that in RC1 soil, and the ammonium nitrogen, nitrate nitrogen, and available potassium content was lower, indicating that fertilization after rotation inhibited the formation of available nitrogen and potassium in soil, thus improving crop quality.

**Table 2.** Soil chemical properties.

| Treatment | EC (μS/cm) | pH | TN (g/kg) | TP (g/kg) | TK (g/kg) | SOC (g/kg) | AN (mg/kg) | AP (mg/kg) | AK (mg/kg) | NN (mg/kg) |
|---|---|---|---|---|---|---|---|---|---|---|
| RC1 | 27.22 ± 1.44 b | 5.6 ± 0.06 c | 0.6 ± 0.002 b | 1.17 ± 0.25 a | 6.66 ± 0.02 c | 76.6 ± 2.3 c | 4.69 ± 0.54 a | 48.4 ± 2.13 c | 76.59 ± 2.5 b | 14.45 ± 0.2 b |
| RC0 | 14.32 ± 0.75 d | 5.98 ± 0.02 a | 0.87 ± 0.004 c | 1.37 ± 0.02 b | 7.16 ± 0.06 a | 78.16 ± 0.93 c | 4.04 ± 0.11 b | 27.82 ± 0.48 c | 109.32 ± 2.65 b | 13.46 ± 0.05 a |
| CC1 | 42.98 ± 4.01 a | 5.49 ± 0.05 d | 0.66 ± 0.02 b | 1 ± 0.12 a | 7.2 ± 0.02 c | 66.99 ± 0.55 c | 3.67 ± 0.05 c | 57.27 ± 0.34 a | 97.01 ± 1.3 b | 16.44 ± 0.2 d |
| CC0 | 12.4 ± 0.99 a | 5.63 ± 0.02 c | 0.51 ± 0.1 a | 0.98 ± 0.09 b | 7.18 ± 0.06 a | 67.41 ± 1.48 c | 2.33 ± 0.41 a | 23.77 ± 0.6 b | 176.29 ± 0.19 a | 13.11 ± 0.13 c |

Note: TN, total nitrogen; TP, total phosphorus; TK, total phosphorus; SOC, soil organic carbon; NN, nitrate nitrogen; AP, available phosphorus; AK, available potassium; and AN, ammonium nitrogen. All data are expressed as the mean ± standard error, *n* = 3. Means within each column and main effects followed by different letters are significantly different ($p < 0.05$) according to Duncan's multiple-range test.

*3.2. Effects of Soybean–Maize Rotation and Fertilization Treatments on Soybean Growth*

Table 3 shows that there was a significant effect of rotation treatment on plant height and stem diameter ($p < 0.05$). The height and stem diameter of plants grown in RC1 soil were decreased by 3.69% and 4.67%, respectively, in comparison to those grown in CC1 soil. Additionally, the height and stem diameter of plants grown in RC0 soil were increased by 19.51% and 6.97%, respectively, in comparison to those grown in CC0 soil. Moreover, there was a very significant effect of fertilization on plant height and stem diameter ($p < 0.01$). The height and stem diameter of plants grown in RC1 soil were increased by 1.39% and 7.49%, respectively, in comparison to those grown in RC0 soil. Additionally, the height and stem diameter of plants grown in CC1 soil were increased by 25.82% and 20.62%, respectively, in comparison to those grown in CC0 soil. Furthermore, the combination of rotation and fertilization had a significant effect on plant height ($p < 0.01$). The height of plants grown in CC0 soil was significantly lower than that of those grown in the other soils.

**Table 3.** Growth indexes of soybean under corn–soybean rotation combined with fertilization.

| Treatment | | Height/cm | Stem Diameter/mm | Number of Nodes |
|---|---|---|---|---|
| Rotation system | RC1 | 91.70 ± 4.16 a | 7.75 ± 0.38 a | 17.17 ± 0.31 a |
| | RC0 | 89.72 ± 1.60 a | 6.42 ± 0.21 b | 16.30 ± 1.00 b |
| Continuous cropping | CC1 | 95.21 ± 2.75 a | 8.13 ± 0.26 a | 17.11 ± 0.91 a |
| | CC0 | 75.67 ± 0.96 b | 6.74 ± 0.17 c | 16.97 ± 0.56 ab |
| Two-factor variance analysis(F) | | | | |
| Cropping pattern | | 8.084 * | 6.208 * | 2.418 |
| Mode of fertilization | | 13.970 ** | 47.391 ** | 2.149 |
| Cropping pattern × Mode of fertilization | | 21.311 ** | 0.059 | 2.938 |

Note: All data are expressed as the mean ± standard error, $n = 3$. Means within each column and main effects followed by different letters are significantly different ($p < 0.05$) according to Duncan's multiple-range test. * and ** represent significance at $p < 0.05$ and $p < 0.01$, respectively. RC1, maize–soybean rotation with regular fertilization; RC0, maize–soybean rotation without fertilization; CC1, soybean continuous cropping with fertilization; and CC0, soybean continuous cropping without fertilization.

*3.3. Effects of Soybean–Maize Rotation and Fertilization Treatments on Soybean Yield and Nitrogen Utilization*

Table 4 shows the very significant effect of the tillage method on the total grain weight ($p < 0.01$). The total weight of grain grown in RC1 soil was increased by 25% in comparison to that of grain grown in CC1 soil. Additionally, the total weight of grain grown in RC0 soil was increased by 2.56% in comparison to that of grain grown in CC0 soil. Moreover, there was a significant effect of fertilization method on the total grain weight ($p < 0.05$). The total weight of grain grown in RC1 soil was increased by 31.25% in comparison to that of grain grown in RC0 soil. Additionally, the total weight of grain grown in CC1 soil was 7.69% higher than that of grain grown in CC0 soil. Furthermore, the tillage method had very significant effects on crop yield and on the agronomic efficiency of nitrogen fertilizer ($p < 0.01$). The crop yield of plants grown in RC1 soil was 12.11% higher than that of plants grown in CC1 soil, and the agronomic efficiency of nitrogen fertilizer was 52.81% lower. Additionally, the crop yield of plants grown in RC0 soil was 21.42% higher than that of plants grown in CC0 soil. Likewise, the fertilization method had a very significant effect on the agronomic efficiency of nitrogen fertilizer ($p < 0.01$), and the effect on crop yield was significant ($p < 0.05$). The crop yield of plants grown in RC1 soil was 9.32% higher than that of plants grown in RC0 soil, and the nitrogen agronomic efficiency decreased by 14.85%. The crop yield of plants grown in CC1 soil was 18.41% higher than that of plants grown in CC0 soil. The combination of rotation and fertilization had a very significant effect on the agronomic efficiency of nitrogen fertilizer ($p < 0.01$): CC1 > RC1.

**Table 4.** Yield components of soybean under different tillage methods combined with fertilization.

| Treatment | | Yield/(kg/ha) | N-Fertilizer Agronomic Efficiency/[kg/(kg·hm²)] | Total Grain Weight/g | 100-Grain Weight/g |
|---|---|---|---|---|---|
| Rotation cropping | RC1 | 2941.18 ± 101.89 a | 10.32 ± 0.36 c | 3.15 ± 0.07 a | 22.00 ± 0.80 a |
| | RC0 | 2725.5 ± 290.17 ab | 12.12 ± 1.29 b | 2.86 ± 0.34 ab | 22.64 ± 0.51 a |
| Continuous cropping | CC1 | 2623.53 ± 92.45 b | 21.87 ± 0.78 a | 2.52 ± 0.31 bc | 22.20 ± 1.78 a |
| | CC0 | 2215.69 ± 97.08 c | 23.57 | 2.34 ± 0.20 c | 19.34 ± 1.61 b |
| Two-factor variance analysis (F) | | | | | |
| Cropping pattern | | 49.838 ** | 2095.038 ** | 135.903 ** | 43.805 ** |
| Mode of fertilization | | 35.397 ** | 135.611 ** | 14.916 ** | 25.396 ** |
| Cropping pattern × Mode of fertilization | | 8.477 * | 30.929 ** | 2.946 | 18.228 * |

Note: All data are expressed as the mean ± standard error, $n = 3$. Means within each column and main effects followed by different letters are significantly different ($p < 0.05$) according to Duncan's multiple-range test. * and ** represent significance at $p < 0.05$ and $p < 0.01$, respectively. RC1, maize–soybean rotation with regular fertilization; RC0, maize–soybean rotation without fertilization; CC1, soybean continuous cropping with fertilization; CC0, soybean continuous cropping without fertilization.

*3.4. Response of Soil Bacterial Community to Soybean–Maize Rotation and Fertilization Treatments*

3.4.1. Effects on Bacterial Diversity

According to Table 5, the coverage index of each treatment was greater than 0.95, indicating that the sequencing results can express most of the microbial community changes. The smaller the Simpson index, the greater the microbial diversity of the sample; the Simpson index of each treatment was less than 0.01, indicating that the microbial diversity of the four samples is higher. Moreover, the Shannon index of each treatment shows various differences. The Shannon index of CC1 was the largest and that of CC0 was the smallest, indicating that the bacterial diversity of CC1 is higher, while that of CC0 is relatively low. The chao1 index of CC0 was higher, and that of RC1 was the lowest, indicating that the bacterial richness following CC0 treatment is relatively high in comparison with that following other treatments, while the richness of RC1-treated bacteria is relatively low.

**Table 5.** The indexes of bacterial alpha diversity.

| Sample | Shannon | Chao1 | Coverage | Simpson |
|---|---|---|---|---|
| RC1 | 6.339251 | 3118.223 | 0.968376 | 0.004401 |
| RC0 | 6.3424 | 3241.878 | 0.96706 | 0.005188 |
| CC1 | 6.360753 | 3220.631 | 0.968456 | 0.004995 |
| CC0 | 6.30414 | 3246.365 | 0.966342 | 0.00628 |

Note: RC1, maize–soybean rotation with regular fertilization; RC0, maize–soybean rotation without fertilization; CC1, soybean continuous cropping with fertilization; CC0, soybean continuous cropping without fertilization.

The effects of soybean–maize rotation and fertilization treatments on soil bacterial community and diversity are shown in Figure 1. The dominant bacteria in soil of all treatments were Candidatus_Solibacter, Bryobacter, Gemmatimonas, Rhodanobacter, Sphingomonas, and Bradyrhizobium; however, their relative abundance in the soil of each treatment was very different. Under rotation, the relative abundance of Bryobacter in RC1 and RC0 soil was higher than that in CC1 (14.55%) and CC0 (15.56%) soil, respectively. Under fertilization, the relative abundance of Bryobacter in RC1 and CC1 soil was lower than that in RC0 (6.15%) and CC0 (5.33%) soil, respectively. The main function of Bryobacter is to decompose lignin and cellulose.

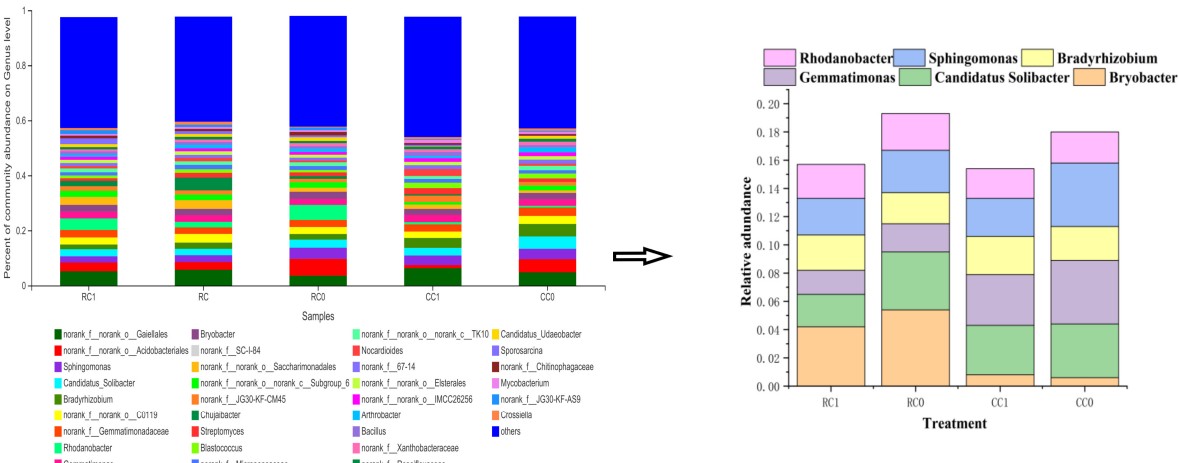

**Figure 1.** Soil bacterial species at the genus level under different treatments.

The relative abundance of Gemmatimonas in RC1, RC0, CC1, and CC0 soil was 2.49%, 2.16%, 2.69%, and 2.44%, respectively. Under rotation, the relative abundance of Gemmatimonas in RC1 and RC0 soil was lower than that in CC1 (7.43%) and CC0 (11.48%) soil, respectively. Under fertilization, the relative abundance of Gemmatimonas in RC1 and CC1 soil was higher than that in RC0 (15.28%) and CC0 (10.25%) soil, respectively. Gemmatimonas has been reported to decompose organic matter in soil, but its existence in a high abundance can reduce plant resistance.

The relative abundance of Bradyrhizobium in RC1, RC0, CC1, and CC0 soil was 1.73%, 2.02%, 3.63%, and 4.52%, respectively. Under rotation, the relative abundance of Bradyrhizobium in RC1 and RC0 soil was lower than that in CC1 (24.79%) and CC0 (55.31%) soil, respectively. Under fertilization, the relative abundance of Bradyrhizobium in RC1 and CC1 soil was lower than that in RC0 (14.36%) and CC0 (19.69%) soil, respectively. Studies have shown that Bradyrhizobium can promote the formation of nodules and fix nitrogen in the air for plant nutrition.

Sphingomonas and Rhodanobacter have good degradation ability. The relative abundance of Sphingomonas in RC1, RC0, CC1, and CC0 soil was 2.27%, 3.8%, 3.5%, and 4.06%, respectively. Under rotation, the relative abundance of Sphingomonas in RC1 and RC0 soil was lower than that in CC1 (35.14%) and CC0 (6.4%) soil, respectively. Under fertilization, the relative abundance of Sphingomonas in RC1 and CC1 soil was lower than that in RC0 (40.26%) and CC0 (13.79%) soil, respectively. The relative abundance of Rhodanobacter in RC1, RC0, CC1, and CC0 soil was 4.17%, 5.46%, 0.62%, and 0.77%, respectively. Under rotation, the relative abundance of Rhodanobacter in RC1 and RC0 soil was higher than that in CC1 (572.58%) and CC0 (609.09%) soil, respectively. Under fertilization, the relative abundance of Rhodanobacter in RC1 and CC1 soil was lower than that in RC0 (23.63%) and CC0 (19.48%) soil, respectively.

In summary, rotation treatment increased the relative abundance of Bryobacter and Rhodanobacter, and inhibited the relative abundance of Candidatus_Solibacter, Gemmatimonas, Bradyrhizobium, and Sphingomonas. Fertilization treatments increased the relative abundance of Gemmatimonas and inhibited the relative abundance of Candidatus_Solibacter, Bryobacter, Bradyrhizobium, Sphingomonas, and Rhodanobacter.

### 3.4.2. Soil-Bacterial-Community-Associated Environmental Factors

Based on the Bray–Curtis coefficient of variation, run 999 times, PCA analysis was used to compare the degree of similarity in the diversity of different samples. Figure 2 shows that the tillage method and fertilization treatment had a significant effect on the soil bacterial community and diversity. Among them, AN, TN, TP, and SOM were positively correlated with RC1 rhizosphere soil, which is consistent with the previous analysis of soil physicochemical indicators. Soil pH and AP were positively correlated with RC0 soil,

indicating that soil rotation without fertilization improved AP accumulation and soil pH, which is in accordance with previous studies [29].

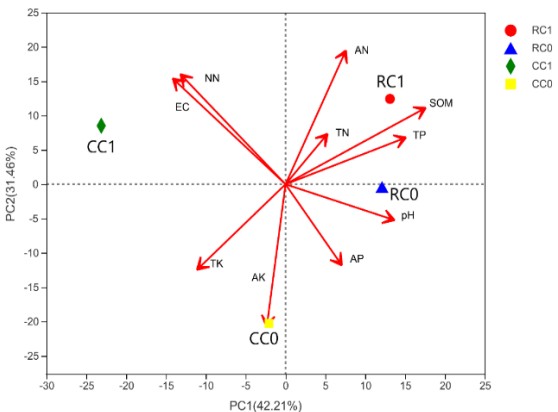

**Figure 2.** Principal component analysis of bacterial community structure.

### 3.4.3. Correlation between Soil Bacterial Community Structure and Soil Environmental Factors

The correlation among the soil microbial community at the genus level, crop yield, and soil physicochemical indexes is shown by the Heatmap in Figure 3. Arthrobacter was significantly positively correlated with available potassium and negatively correlated with EC. Bacillus and Gemmatimonas were significantly positively correlated with EC and negatively correlated with pH. Organic matter and total potassium were significantly positively correlated with Mycobacterium and negatively correlated with Sporosarcina. Total phosphorus content was significantly negatively correlated with Blastococcus and Bradyrhizobium, and significantly positively correlated with Rhodanobacter. Ammonium nitrogen was significantly positively correlated with Chujaibacter and crop yield was negatively correlated with Sporosarcina.

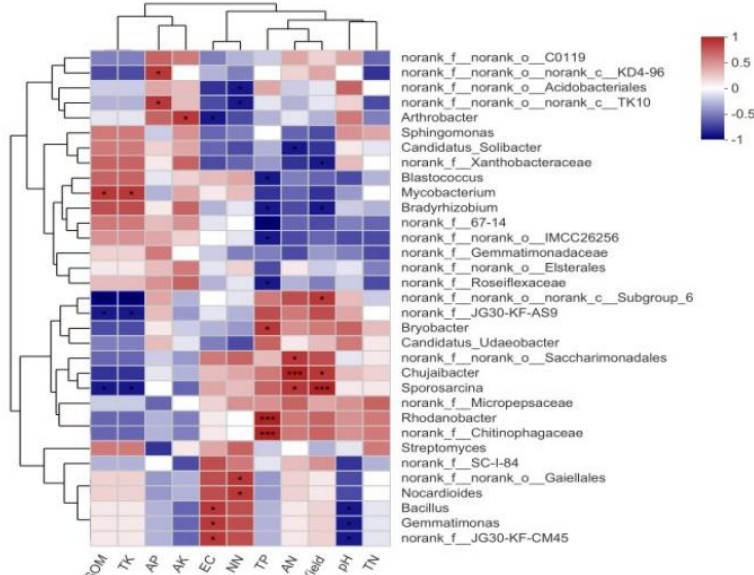

**Figure 3.** Heatmap of soil bacterial community at the genus level and soil environmental factors. The *x*-axis and *y*-axis are environmental factors and species, respectively. The correlation R value and *p* value were obtained by calculation. The R value is shown in different colors in the graph, and *p* values less than 0.05 are marked by *. The legend on the right is the color interval of different R values; * $0.01 < p \leq 0.05$, *** $p \leq 0.001$.

### 3.4.4. Molecular Ecological Network Structure

In nature, there are generally complex interactions among microorganisms, and these interactions are an important characteristic of microbial communities. Therefore, it is important to study the synergistic mechanism to reveal the structure and function of microbial communities. To explore the relationship between the bacterial species in the soil and the processes of rotation and fertilization, soil bacteria were subjected to high-throughput sequencing. The top 100 most abundant genera were selected, and molecular networks were constructed for RC1, RC0, CC1, and CC0 soil. As shown in Figure 4, the molecular networks of RC1, RC0, CC1, and CC0 soil had 3508, 3730, 3306, and 3354 connections, respectively (Figure 5a). The number of negative correlation lines in the bacterial networks of RC1, RC0, CC1, and CC0 soil was 1619, 1573, 2262, and 1895, respectively. The proportion of the corresponding total number of connections was 47.86%, 42.17%, 68.42%, and 56.5%, respectively, indicating that the proportion of competitive relationships among bacterial species was very different in the four soils. Sphingomonas had the largest number of connections, followed by Candidatus_Solibacter, Bradyrhizobium, Gemmatimonas, Rhodanobacter, and Bryobacter. The treatment centrality of RC1, RC0, CC1, and CC0 soil was 0.39, 0.4, 0.37, and 0.36, respectively (Figure 5b); thus, rotation effectively increased the stability of the soil bacterial network structure.

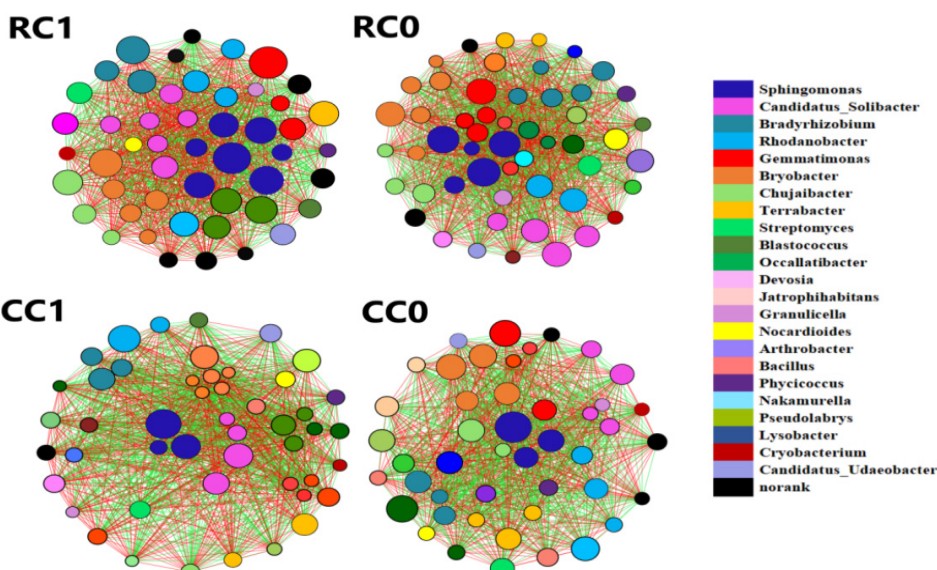

**Figure 4.** Molecular ecological networks of bacterial communities at the genus level. Each node signifies an OTU, which corresponds to a bacterial species. Colors of the nodes represent different dominant bacterial genera. Red lines represent positive interactions between two individual nodes, while green lines represent negative interactions.

The number of bacterial genus connections in the network was different in each soil. As shown in Figure 5c, the number of connections for Sphingomonas and Rhodanobacter increased after rotation and decreased after fertilization. Additionally, the number of connections for Candidatus_Solibacter, Bradyrhizobium, and Gemmatimonas were higher after rotation than in continuous cropping. Moreover, the number of connections for Candidatus_Solibacter and Gemmatimonas decreased after fertilization; however, for Bradyrhizobium they were increased. Furthermore, the number of connections for Bryobacter in CC1 and CC0 soil was 19 and 17, respectively, which is significantly less than 48 and 58 in RC1 and RC0 soil, respectively. In summary, crop rotation increased the relative abundance of Sphingomonas, Rhodanobacter, Bryobacter, and other beneficial bacteria. The bacterial ecological network structure of the four soils was significantly different. The competitive relationship among species in the bacterial ecological network was strongest in

CC1 soil, while the symbiotic relationship among species in the bacterial ecological network was strongest in RC0 soil.

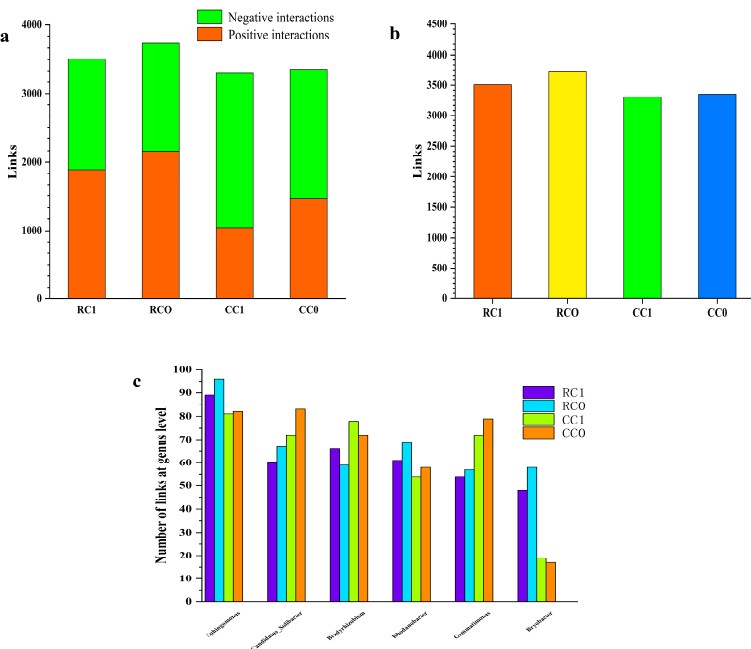

**Figure 5.** Number of network links (**a**), degree of centrality (**b**), and number of links between advantageous bacteria at the genus level (**c**).

### 3.4.5. Molecular Ecological Network Structure

According to the PICRUSt gene prediction, nitrogen-metabolism-related enzymes, and gene expression abundance in different soil samples, were selected from the gene function annotation data obtained by metagenomic sequencing, and the abundance was analyzed (Table 6). There were differences in the gene expression levels of nitrogen metabolic enzymes among the different soils. The gene expression levels of enzymes involved in denitrification were 49.59% higher than those in crop rotation systems. The expression levels of enzymes involved in nitrification and ammonification were higher than those in continuous cropping, by 32.69% and 25.44%, respectively. The expression levels of nitrogenases involved in nitrogen fixation were 45.44% higher than those in continuous cropping. These data indicate that rotation reduced soil denitrification, as well as NO and $N_2O$ emissions, thereby reducing the loss of nitrogen from the soil. Moreover, rotation enhanced soil ammonification, nitrification, and nitrogen fixation, in addition to increasing soil $NO_3^-$-N and $NH_4^+$-N content [30–32].

**Table 6.** Gene expression levels of nitrogen-related metabolic enzymes in the different soils.

| Metabolic Pathway | Enzyme Number | Enzyme | RC1 | RC0 | CC1 | CC0 |
|---|---|---|---|---|---|---|
| | 1.7.2.6 | Hydroxylamine reductase | 99.50 | 63.00 | 69.00 | 85.00 |
| Nitrification | 1.14.18.3 | Methane monooxygenase | 73.00 | 70.00 | 59.00 | 58.00 |
| | 1.14.99.39 | Ammonia monooxygenase | 73.00 | 70.00 | 59.00 | 58.00 |
| | 1.7.2.5 | Nitric oxide reductase | 1241.33 | 898.65 | 2195.47 | 2564.67 |

**Table 6.** *Cont.*

| Metabolic Pathway | Enzyme Number | Enzyme | RC1 | RC0 | CC1 | CC0 |
|---|---|---|---|---|---|---|
| Denitrification | 1.7.2.1 | Nitrite reductase | 2421.03 | 2099.65 | 3157.49 | 3542.19 |
| | 1.7.99.1 | Hydroxylamine reductase | 649.03 | 649.43 | 769.32 | 664.72 |
| | 1.7.2.4 | Nitrous oxide reductase | 1270.53 | 1195.15 | 1502.66 | 1662.69 |
| | 1.7.7.2 | Nitrate reductase | 706.65 | 597.82 | 695.66 | 790.99 |
| | 3.5.5.1 | Nitrilase | 2242.52 | 2212.55 | 1845.52 | 1847.66 |
| Ammoniation | 1.4.1.4 | Glutamate dehydrogenase | 3701.32 | 3346.57 | 3703.46 | 3292.57 |
| | 1.4.1.2 | Glutamate dehydrogenase | 11,380.13 | 8804.01 | 8569.94 | 9067.96 |
| | 2.7.2.2 | Carbamate kinase | 2353.5 | 1761.50 | 1325.72 | 1261.13 |
| | 3.5.1.49 | Formamidase | 3708.94 | 3123.03 | 2036.37 | 2043.31 |
| | 4.2.1.104 | Hydrogenase | 2813.80 | 2884.13 | 1729.50 | 1805.40 |
| | 1.18.6.1 | Nitrogenase | 2846.84 | 3032.04 | 1438.33 | 1660.12 |
| Nitrogen fixation | | | | | | |

Note: RC1, maize–soybean rotation with regular fertilization; RC0, maize–soybean rotation without fertilization; CC1, soybean continuous cropping with fertilization; CC0, soybean continuous cropping without fertilization.

*3.5. Effects of Soybean–Maize Rotation and Fertilization on Soil Fungal Community*

3.5.1. Fungal Community Diversity in Soil

According to Table 7, the coverage index of each treatment was greater than 0.99, indicating that the sequencing results could express most of the fungal community changes. The Simpson index was greater than 0.01 but less than 0.1, indicating that the fungal diversity of the four samples was higher, but that the diversity of fungi was less than that of bacteria. The Shannon index of RC0 was the largest and that of CC1 was the smallest. Moreover, the Chao1 index of RC0 was also higher and that of CC1 was the lowest, indicating that the diversity and richness of fungi following RC0 treatment were relatively high in comparison with other treatments, while the diversity and richness of fungi treated with CC1 were relatively low.

**Table 7.** The indexes of fungal alpha diversity.

| Sample | Shannon | Chao1 | Coverage | Simpson |
|---|---|---|---|---|
| RC1 | 3.275589 | 252.5172 | 0.999179 | 0.077116 |
| RC0 | 3.528299 | 299.3871 | 0.998921 | 0.056446 |
| CC1 | 3.249732 | 233.3226 | 0.999155 | 0.075076 |
| CC0 | 3.495963 | 267.75 | 0.999179 | 0.057651 |

Note: RC1, maize–soybean rotation with regular fertilization; RC0, maize–soybean rotation without fertilization; CC1, soybean continuous cropping with fertilization; CC0, soybean continuous cropping without fertilization.

The effects of soybean–maize rotation and fertilization on soil fungal community and diversity are shown in Figure 6. The dominant fungal genera in all treatments were Mrakia, Heterocephalacria, Tolypocladium, Fusarium, and Chaetomium; however, the relative abundance in each treatment varied greatly. The relative abundance of Mrakia in RC1, RC0, CC1, and CC0 soil was 10.2%, 6.1%, 5.9%, and 0.99%, respectively. Under rotation, the relative abundance of Mrakia in RC1 and RC0 soil was higher than that in CC1 (72.88%) and CC0 (516.16%) soil, respectively. Under fertilization, the relative abundance of Mrakia in RC1 and CC1 soil was higher than that in RC0 (67.21%) and CC0 (495.96%) soil, respectively. Previous studies have confirmed that Mrakia can loosen soil, as well as produce a large number of antibiotics and induce a variety of disease resistances to effectively increase crop yield and quality [33].

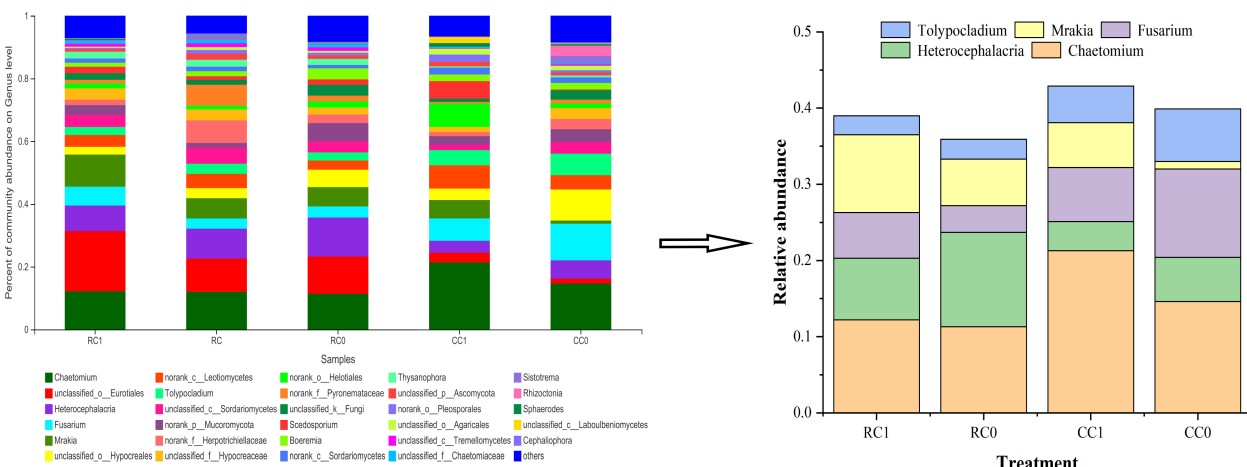

**Figure 6.** Fungal species at the genus level under different treatments.

The relative abundance of Heterocephalacria in RC1, RC0, CC1, and CC0 soil was 8.1%, 12.4%, 3.8%, and 5.8%, respectively. Under rotation, the relative abundance of Heterocephalacria in RC1 and RC0 soil was higher than that in CC1 (161.29%) and CC0 (113.79%) soil, respectively. Under fertilization, the relative abundance of Heterocephalacria in RC1 and CC1 soil was lower than that in RC0 (34.68%) and CC0 (34.48%) soil, respectively. Previous studies have demonstrated that Heterocricephalcria decomposes cellulose, lignin, and pectin in soil to improve nutrient status.

The relative abundance of Tolypocladium in RC1, RC0, CC1, and CC0 soil was 2.5%, 2.6%, 4.8%, and 6.9%, respectively. Under rotation, the relative abundance of Tolypocladium in RC1 and RC0 soil was lower than that in CC1 (47.91%) and CC0 (62.32%) soil, respectively. Under fertilization, the relative abundance of Tolypocladium in RC1 and CC1 soil was higher than that in RC0 (4%) and CC0 (43.75%) soil, respectively.

The relative abundance of Fusarium in RC1, RC0, CC1, and CC0 soil was 6%, 3.5%, 7.1%, and 11.6%, respectively. Under rotation, the relative abundance of Fusarium in RC1 and RC0 soil was lower than that in CC1 (15.49%) and CC0 (69.83%) soil, respectively. Under fertilization, the relative abundance of Fusarium in RC1 soil was higher than that in RC0 (71.43%) soil, and the relative abundance of Fusarium in CC1 soil was lower than that in CC0 (38.79%) soil. Previous studies have shown that Fusarium can infect a variety of economically important crops, causing soybean Fusarium root rot [34,35], Soybean soreshin blight [36], and Sclerotinia sclerotiorum (Lib.) de Bary [37].

The relative abundance of Chaetomium in RC1, RC0, CC1, and CC0 soil was 12.2%, 11.3%, 21.3%, and 14.6%, respectively. Under rotation, the relative abundance of Chaetomium in RC1 and RC0 soil was lower than that in CC1 (42.72%) and CC0 (22.6%) soil, respectively. Under fertilization, the relative abundance of Chaetomium in RC1 and CC1 soil was higher than that in RC0 (7.96%) and CC0 (45.89%) soil, respectively. Previous studies have indicated that Chaetomium can effectively degrade cellulose and organic matter as well as protect against plant pathogens. In summary, the relative abundance of yeast Mrakia and Heterocephalacria was increased under rotation, and the relative abundance of Heterocephalacria was decreased under fertilization. Furthermore, fertilization after crop rotation increased the abundance of Fusarium.

### 3.5.2. Fungal Species Composition

Figure 7 demonstrates that rotation and fertilization had significant effects on soil fungal community and diversity. There was a strong positive correlation among AN, TN, and RC1 rhizosphere soil, indicating that the increase in soil nitrogen content after rotation was a key factor affecting the diversity of the soil fungal community. There was a positive correlation among pH, TP, AP, SOM, and RC0 soil, indicating that the absence of fertilization after rotation was more conducive to an improvement in soil pH, organic

matter, and phosphorus accumulation. EC and NN were positively correlated with CC1 soil, and TK and AK were positively correlated with CC0 soil, indicating that the concentration of salt ions in soil increased after continuous cropping of soybean; and the absence of fertilization treatment was more conducive to the accumulation of potassium in soil.

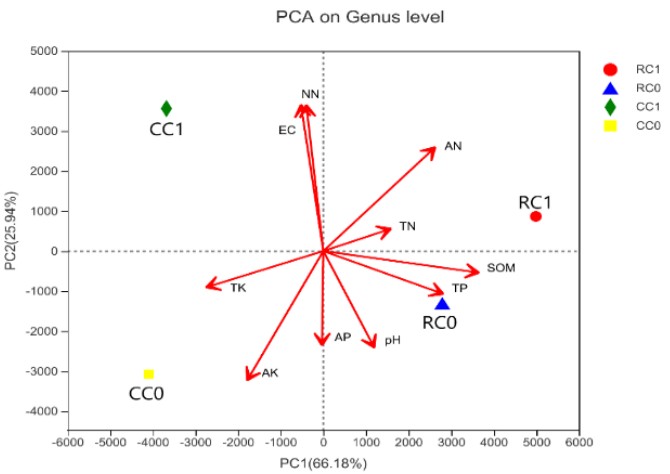

**Figure 7.** Principal component analysis of fungal community structure.

### 3.5.3. Correlation between Soil Fungal Community Diversity and Soil Environmental Factors

The relationship between fungal community diversity and environmental factors indicates that Mrakia was significantly positively correlated with ammonium nitrogen content, extremely significantly positively correlated with crop yield, and significantly negatively correlated with organic matter and total potassium content (Figure 8). Additionally, Heterocephalacria was significantly positively correlated with pH, and Chaetomium was significantly negatively correlated with pH. Moreover, Chaetomium and Tolypocladium, which are harmful to soybean growth, were significantly negatively correlated with crop yield and ammonium nitrogen content.

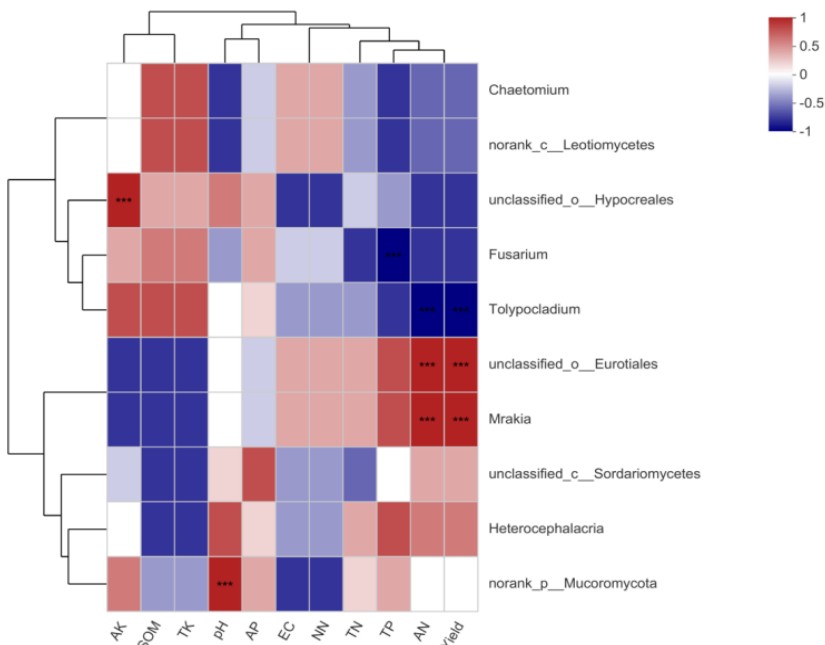

**Figure 8.** Heatmap of soil fungal community at the genus level and soil environmental factors. Note: *** $p \leq 0.001$.

### 3.5.4. Molecular Ecological Network Structure of Fungal Community Diversity

To explore the relationship between the fungal species in the soil and the processes of rotation and fertilization, soil fungi were subjected to high-throughput sequencing. The top 100 most abundant genera were selected, and molecular networks were constructed for RC1, RC0, CC1, and CC0 soil (Figure 9). Overall, RC1, RC0, CC1, and CC0 soil had 3528, 3188, 3794, and 3289 connections, respectively (Figure 10a). The number of negative correlation lines in the fungal network of RC1, RC0, CC1, and CC0 soil was 1928, 2155, 1757, and 1460, respectively. The proportion of the corresponding total number of connections was 54.65%, 67.59%, 46.3%, and 44.39%, respectively, indicating that the proportion of competitive relationships among fungal species in the four soils was very different. Chaetomium had the most connections, followed by Fusarium, Heterocephalacria, Mrakia, and Tolypocladium. The treatment centrality of RC1, RC0, CC1, and CC0 soil was 0.36, 0.33, 0.39, and 0.33, respectively (Figure 10b); thus, rotation reduced the stability of the soil fungal network structure.

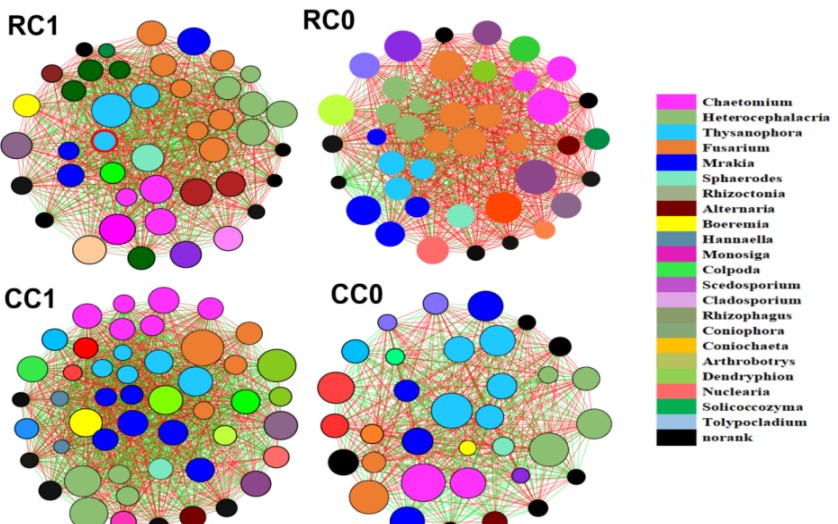

**Figure 9.** Molecular ecological networks of fungal communities at the genus level.

The number of fungal species connections in the four networks was different in each soil. As shown in Figure 10c, Chaetomium and Fusarium showed a decrease in the number of connections after rotation and an increase in the number of connections after fertilization. The number of connections for Tolypocladium after rotation was lower than that in continuous cropping, and the number of connections was decreased after fertilization. The number of connections for Heterocephalacria and Mrakia was increased after rotation and fertilization. It is worth noting that the number of connections for Heterocephalacria in CC1 and CC0 soil was 27 and 18, respectively, which was significantly smaller than that in RC1 and RC0 soil (54 and 69, respectively). In summary, there were differences among the fungal ecological network structures of the four soils. The competition among species in the fungal ecological network was strongest in RC0 soil, while the symbiotic relationship among species in the fungal ecological network was strongest in CC1 soil. The addition of chemical fertilizer increased the relative abundance of potential plant pathogens such as Fusarium. Rotation treatment not only increased the abundance of beneficial bacteria such as Mrakia, but also increased soil nutrients to protect against plant pathogens.

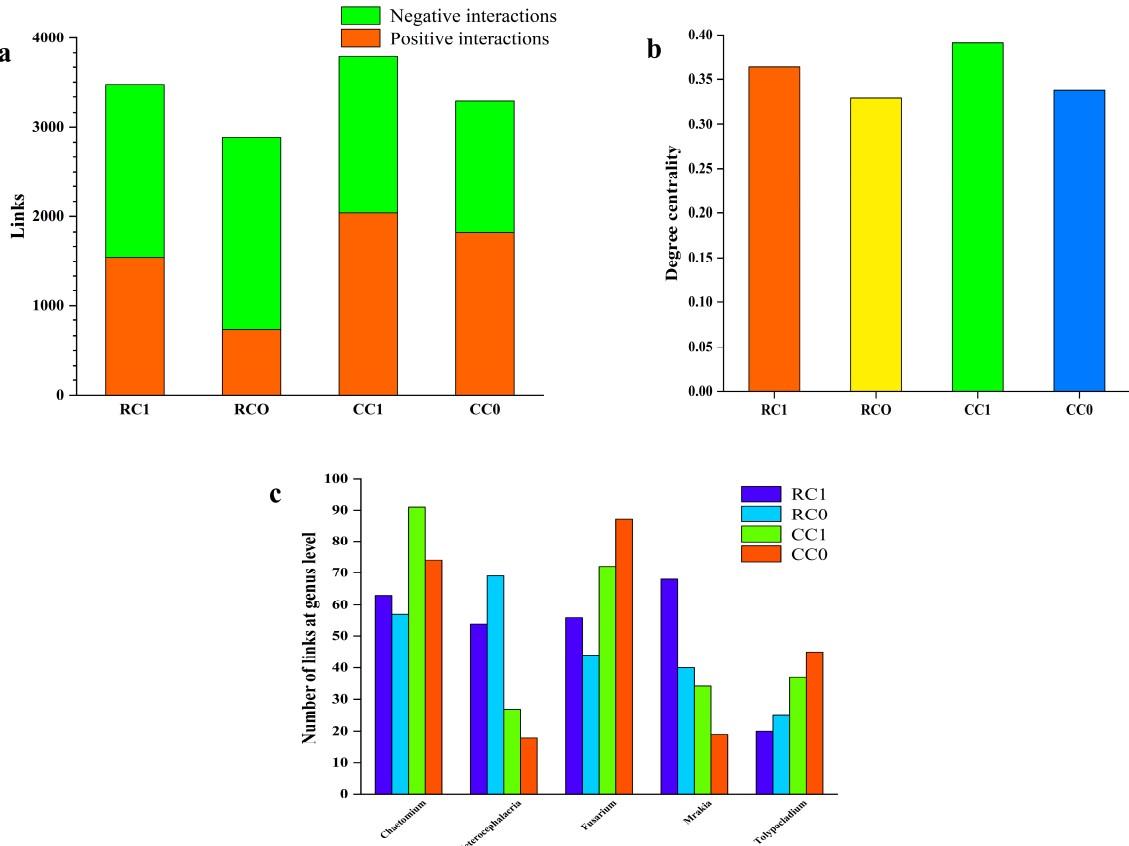

**Figure 10.** Number of network links (**a**), degree centrality (**b**), and number of links between advantageous fungi at the genus level (**c**).

### 3.5.5. FUNGuild Prediction of Soil Fungal Genes

According to FUNGuild gene prediction, the expression levels of saprophytic and pathogenic fungal genes in soil samples under rotation and fertilization treatments were evaluated (Figure 11). The relative abundance of saprophytic fungal genes in RC1, RC0, CC1, and CC0 soil was 1.13%, 1.83%, 0.56%, and 1.21%, respectively. Under rotation, the relative abundance of saprophytic fungal genes in RC1 and RC0 soil was higher than that in CC1 (101.79%) and CC0 (51.24%) soil, respectively. Under fertilization, the relative abundance of saprophytic fungal genes in RC1 and CC1 soil was lower than that in RC0 (38.25%) and CC0 (53.72%) soil, respectively. These data indicate that Dung/Wood saprotrophs increased the content of organic matter in soil, which was the main reason for the increase in soil nutrients after rotation. The relative abundance of pathogenic fungal genes in RC1, RC0, CC1, and CC0 soil was 0.79%, 1.63%, 0.86%, and 0.95%, respectively. Under rotation, the relative abundance of pathogenic fungal genes in RC1 soil was lower than that in CC1 (8.14%) soil, and the relative abundance of pathogenic fungal genes in RC0 soil was higher than that in CC0 (71.59%) soil. The relative abundance of pathogenic fungal genes in RC1 and CC1 soil was lower than that in RC0 (51.53%) and CC0 (9.47%) soil, respectively. Rotation increased the relative abundance of Dung/Wood saprotrophs, and fertilization treatment after rotation further increased their relative abundance.

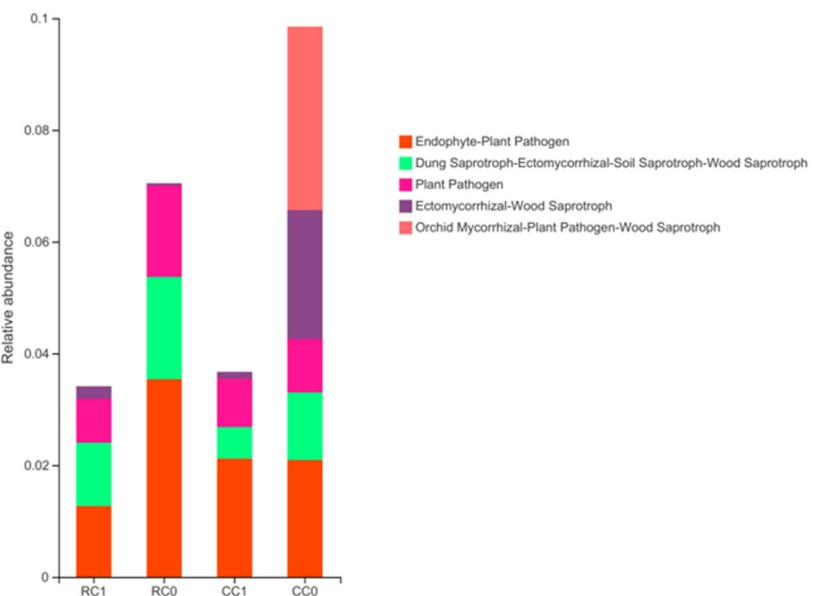

**Figure 11.** FUNGuild gene prediction of soil fungi.

## 4. Discussion

Soybean is an important source of food and oil in China; therefore, it is imperative to achieve a high crop yield and protect soil fertility. The ecological and environmental problems caused by excessive fertilization have attracted increasing attention [38,39]. In the present study, we explored the ecological associations among soil physicochemical properties, soybean yield and agronomic traits, and soil microbial community and function using different non-fertilization methods under the 10-year soybean–maize rotation cropping system. We found that soybean agronomic traits were closely related to crop yield, which is consistent with previous reports demonstrating that the ratio of main stem length to stem diameter is an important indirect indicator of soybean yield [40] and that there is a significant positive correlation between soybean stem diameter and yield per plant [41].

Here, there was a significant effect of rotation on plant height and stem diameter ($p < 0.05$). The height and stem diameter of plants grown in RC0 soil were increased by 19.51% and 6.97%, respectively, in comparison with those of plants grown in CC0 soil. Additionally, there was a very significant effect of fertilization on plant height and stem diameter ($p < 0.01$). The height and stem diameter of plants grown in RC1 soil were increased by 1.39% and 7.49%, respectively, in comparison with those of plants grown in RC0 soil. The height and stem diameter of plants grown in CC1 soil were increased by 25.82% and 20.62%, respectively, in comparison with those of plants grown in CC0 soil. Moreover, the tillage method had an extremely significant effect on crop yield ($p < 0.01$). The crop yield of plants grown in RC1 soil was 12.11% higher than that of plants grown in CC1 soil, and the crop yield of plants grown in RC0 soil was 21.42% higher than that of plants grown in CC0 soil. Furthermore, fertilization had a very significant effect on the agronomic efficiency of nitrogen fertilizer ($p < 0.01$) and a significant effect on crop yield ($p < 0.05$). The crop yield of plants grown in RC1 soil was 9.32% higher than that of plants grown in RC0 soil, and the crop yield of plants grown in CC1 was increased by 18.41% in comparison with that of plants grown in CC0 soil, indicating that rotation and fertilization increased crop yield. The nitrogen agronomic efficiency of soybean continuous cropping was higher under fertilization than under rotation. Due to biological nitrogen fixation by soybean itself, the transformation and maintenance of soil's available nitrogen content are of great significance for successful nitrogen absorption [10]. Under rotation, although the crop yield in the absence of fertilization is lower than that with normal fertilization, the agronomic efficiency of nitrogen fertilizer in the absence of fertilization is higher than that with normal fertilization. Due to the high nitrogen application rate of the current crop, succeeding crops grown in the absence of additional nitrogen fertilizer also display efficient

nitrogen absorption and an improved utilization rate of residual nitrogen fertilizer [42]. Soil nitrate nitrogen is the available nitrogen that can be directly absorbed and utilized by plants and is an important indicator of soil nitrogen level. The amount of soil nitrate nitrogen residue depends mainly on the absorption of soil nitrate nitrogen by crop roots, and changes in soil nitrate nitrogen content significantly affect the nitrogen content of plants [43]. Our study demonstrates that rotation and fertilization, and the combination of tillage and fertilization, had very significant effects on nitrate nitrogen content during the crop maturation period ($p < 0.01$). Principal component analysis (PCA) was used to comprehensively analyze the differences in bacterial and fungal community composition among the soil samples under different treatments. The results indicate that nitrate nitrogen content affected soil microbial community structure. In terms of leguminous plants and productivity, it is necessary to further understand how nitrate nitrogen affects crop yield to improve cultivation measures, yield, and soil fertility.

Microbial communities have an important impact on plant health and growth [44]. Leguminous plants play a key role in the integrated management of soil fertility, and the symbiotic relationship between leguminous crops and rhizobia can fix $N_2$, which provides organic resources. Moreover, other limitations such as competition between crops and weeds can be offset by enhancing fertilizer uptake and inhibiting weeds [45]. Rhizobia can promote the formation of nodules and provide nitrogen sources for crops, while crops can provide carbon sources for rhizobia. The enrichment of rhizobia can also inhibit some pathogens [15]. The results show that rhizobia caused an increase in all 13 functional bacteria within the nitrogen cycle, which may be related to the number of beneficial functional bacteria in prokaryotic microbial communities in the rhizosphere, and rhizobia favor the strengthening of nitrogen cycle function [46]. Studies have shown that the reasonable application of nitrogen fertilizer can promote nodule formation and improve the nitrogen fixation capacity of soybean; however, when the exogenous nitrogen level is high, nitrogen will exert a negative effect. An excessive application of nitrogen fertilizer inhibits nodule formation, resulting in a significant reduction in the number of nodules and a decrease in soybean yield [47]. An appropriate reduction in nitrogen application during soybean cultivation can significantly enhance the nitrogen fixation capacity of nodules, increase dry matter accumulation, and effectively increase the number of pods per plant, as well as the number of grains per plant, thereby increasing crop yield [48]. A prediction of bacterial functional genes indicates that enzymes related to ammonification and nitrification were higher in soil during continuous cropping than under rotation. Under the action of soil nitrogen-cycling microorganisms, nitrogen-fixing microorganisms convert $N_2$ from the biosphere into ammonia nitrogen by nitrogen fixation. Furthermore, nitrogen-fixing microorganisms convert ammonia nitrogen into organic or nitrate nitrogen by ammonification and nitrification. Finally, organic and nitrate nitrogen are converted into $N_2$ or NO by denitrification [49,50]. The present study also confirmed a high agronomic efficiency of continuous cropping with nitrogen fertilizer and rotation. The relative abundance of Bradyrhizobium in RC0 (2.02) soil was higher than that in RC1 (1.73) soil under rotation. However, the crop yield of plants grown in RC1 soil was higher than that of plants grown in RC0 soil, but this did not lead to a reduction in crop yield. Although legume rotation can improve the soil environment and increase the nitrogen fixation capacity, fertilization is still the main reason for the large increase in crop yield. Bradyrhizobium was significantly negatively correlated with total grain weight and yield. Due to nitrogen fixation by soybean, continuous cropping resulted in an increase in Bradyrhizobium, the high abundance of which reduced crop yield. According to the predicted bacterial functional genes, the abundance of nitrogenases in continuous cropping was 45.44% higher than that in rotation. Additionally, the abundance of nitrogenases was also higher in RC0 soil than in RC1 soil, which is consistent with the abundance of Bradyrhizobium. This finding indicates that rotation changed the abundance of microorganisms related to nitrogen transformation in soil. Fusarium is a large and complex fungal genus that includes many plant pathogens, such as Fusarium oxysporum and Fusarium equiseti. These pathogens can lead to root

rot disease in soybean. The increase in Fusarium abundance suggests that soybean continuous cropping may increase the occurrence of disease [10]. Studies have shown that soybean–maize rotation increases the yield by 74% in comparison with monoculture, and that soybean yield is significantly negatively correlated with disease incidence ($-0.90$) [24]. In our experiment, the relative abundance of Fusarium under rotation was lower than that in continuous cropping. The relative abundance of Fusarium in RC1 soil was 47.32% lower than that in CC1 soil, and the relative abundance of Fusarium in RC0 soil was 69.55% lower than that in CC0 soil. Fusarium was significantly negatively correlated with 100-grain weight. These results are similar to those previously reported [46]. It is worth noting that the relative abundance of Fusarium in RC1 soil was higher than that in RC0 soil under maize–soybean rotation, indicating that the application of fertilizer resulted in an increase in Fusarium abundance. Fertilization increased the diversity of fungal pathogens and the risk of disease occurrence during soybean production.

Analysis of the bacterial and fungal ecological networks demonstrated that rotation significantly affected the community and diversity of bacteria and fungi, which was very different among the soils. Changes in soil physicochemical properties play a leading role in the network structure. It has been reported that the bacterial community is strongly affected by soil pH and that the optimum growth pH range of most bacteria is narrow [51]. The significant changes in soil pH under different fertilization treatments may lead to the instability of the bacterial network structure. At the same time, the fungal network can transmit soil environmental changes to the entire network over a very short period of time, resulting in an unstable network structure [52]. The number of network connections for Sphingomonas, Rhodanobacter, and Bryobacter in RC1 soil was significantly higher than that in CC1 soil, which is in accordance with previous data [53]. The number of network connections for Chaetomium was found to be the highest among the fungi, which is consistent with its identity as the key species of the fungal network in paddy soils in eastern Asia [54].

## 5. Conclusions

Our data explain the theoretical causes, the effects of changes in metabolic function from a microscopic perspective, and reveal the number and structure of soil microbial communities. These results help us to more accurately describe the mechanism underlying functional soil changes, which is of great significance in improving the stability of agricultural ecosystems and rational scientific fertilization. Both rotation and fertilization affected crop yield and soil microbial community. Rotation increased soybean yield in the presence or absence of fertilization, and a combination of rotation and fertilization had a significant effect on nitrogen agronomic efficiency. Rotation can increase the community diversity index of soil bacteria and fungi, as well as increase the relative abundance of certain beneficial bacteria such as Sphingomonas, Rhodanobacter, and Bryobacter, while reducing the relative abundance of plant pathogenic fungi, such as Fusarium.

**Author Contributions:** H.Z. and W.Z. designed the experiment. Y.W. and L.Z. wrote the first manuscript and performed all of the statistical analyses. Y.W., L.Z., F.M., Z.L., X.A. and X.J. collected the data in the field and lab. All authors have read and agreed to the published version of the manuscript.

**Funding:** This research was supported by the China Agriculture Research System of MOF and MARA (No. CARS-04-PS14), the Young and Middle-aged Scientific and Technological Innovation and Entrepreneurship Outstanding Talent (team) Project (20210509012RQ), and the Department of Key R&D Project of Jilin Province Science and Technology (No. 20200402040NC).

**Data Availability Statement:** The datasets generated and analyzed during the current study are available from the corresponding author on reasonable request.

**Conflicts of Interest:** The authors declare that the research was conducted in the absence of any commercial or financial relationships.

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
