# Peer review of "Responses of Soil Microbial Communities in Soybean–Maize Rotation to Different Fertilization Treatments"

_agronomy, doi:10.3390/agronomy13061590_

Round 1

Reviewer 1 Report

Soil microbes involves in nutrient cycles and play a crucial role in ecological functioning of the ecosystem. Vegetation types and fertilization input could remarkably alter the microbial abundance and community composition. Therefore, it is important to understanding how microbial abundance and community operate in ecological and biogeochemical processes under different agriculture managements. The authors investigated the responses of soil microbial communities to soybean-maize rotation and fertilization treatments and obtained some valuable data and results. The results is essential to understand the mechanisms underlying microbe-driven changes in soil nutrient cycling in farmland. The topic of the study fits the scope of the journal. The interpretation of the results is appropriate. However, the authors should highlight the novel of the paper, provide more information in relation to the methods. English writing/clarity needs additional attention. Authors should pay more efforts for careful modification of their manuscript.

Detail comments:

1. I suggest the authors revise the paper title as “Responses of soil microbial communities to soybean-maize rotation and different fertilization treatments”. 

2. Line 115, m2, 2 should be used in superscript.

3. Materials and Methods section, the authors should describe the 2.5 Microbiological analysis and 2.6. Data analysis in more detail.

4. In the text P < 0.05, P is not used in italic but it is used in italic under the table 2 and table 3.

5. Line 260-266, Soil pH and AP were positively correlated with RC0 soil, indicating that soil rotation without fertilization improved AP accumulation and soil pH, which is in accordance with previous studies[29]........ EC and NN were positively correlated with CC1 soil, TK and AK were positively correlated with CC0 soil. which is in agreement with previous reports that salt content had significant effects on yield [31]. These sentences should be removed to the discussion section.

6. Line 320-321, .3.8. Effect of total amount of straw return with N fertilizer transport on C and N fractions and N accumulation in rice fields under different soil type conditions?

7. Line 331, N2O, line 333, NO3-N and Line 334 NH4+-N, 2, 3, and 4 should be used in subscript.

8. The meaning of RC1, RC0, CC1, and CC0 should be given in the table and figure titles.

9. Line 395, 419, soil fungal microbial community, microbial should be deleted.

10. Line 487-499, these sentences are the results descriptions, but not the discussion.

11. Line 524-534, the authors described more about the nodule, their relationships with soil microbe should be discussed.

12. The indicative significance of the results and outlook should be given in the conclusions section.

English writing needs improvement.

Author Response

Detail comments:

  1. I suggest the authors revise the paper title as “Responses of soil microbial communities to soybean-maize rotation and different fertilization treatments”.

The title has been changed as suggested.

  1. Line 115, m2, 2 should be used in superscript.

This section has been modified.

  1. Materials and Methods section, the authors should describe the 2.5 Microbiological analysis and 2.6. Data analysis in more detail.

Details of the corresponding data analysis and trial processing have been added.

  1. In the text P < 0.05, P is not used in italic but it is used in italic under the table 2 and table 3.

This issue has been fixed.

  1. Line 260-266, Soil pH and AP were positively correlated with RC0 soil, indicating that soil rotation without fertilization improved AP accumulation and soil pH, which is in accordance with previous studies[29]........ EC and NN were positively correlated with CC1 soil, TK and AK were positively correlated with CC0 soil. which is in agreement with previous reports that salt content had significant effects on yield [31]. These sentences should be removed to the discussion section.

This section has been removed.

  1. Line 320-321, .3.8. Effect of total amount of straw return with N fertilizer transport on C and N fractions and N accumulation in rice fields under different soil type conditions?

This was a typo and has been removed.

  1. Line 331, N2O, line 333, NO3-N and Line 334 NH4+-N, 2, 3, and 4 should be used in subscript.

This issue has been revised.

  1. The meaning of RC1, RC0, CC1, and CC0 should be given in the table and figure titles.

The corresponding information has been added.

  1. Line 395, 419, soil fungal microbial community, microbial should be deleted.

This section has been removed.

  1. Line 487-499, these sentences are the results descriptions, but not the discussion.

These sentences have been rearranged.

  1. Line 524-534, the authors described more about the nodule, their relationships with soil microbe should be discussed.

Thank you for your advice. The interrelationship between rhizobia and soil microorganisms and its causes have been added.

  1. The indicative significance of the results and outlook should be given in the conclusions section.

Thank you for your advice. This has been changed and highlighted in yellow.

Reviewer 2 Report

Soil is a key component of the agroecosystem. The intensification of agriculture to meet the growing needs of society can lead to the deterioration, depletion and degradation of the soil cover. Crop rotation is a key method for managing soil nutrients, coordinating plant nutrient uptake and nutrient balance in the soil. Crop rotation maintains soil health, increases soil microbial diversity, and improves soil microbial activity. Soil microorganisms can regulate soil microecology, promote mineral and nutrient cycling, and contribute to soil nutrient diversity.

This paper evaluates a 10-year soybean-corn rotation system with and without various fertilization practices to explore the ecological relationships between soil properties, soybean yield, agronomic traits, microbial communities and functions.

The undoubted advantage of the work is the assessment of the diversity of not only bacteria, but also fungi, as well as the construction of structures of molecular ecological network structure.

However, there are a few remarks about the work:

1. Perhaps it is necessary to calculate and present the indices of alpha diversity of microbial and fungal communities (Shannon, Chao1, etc.).

2. The methods used for data analysis should be more fully described: how analyzes were carried out for the analysis of principal components, correlations, molecular ecological network structure (sections 3.4.2, 3.4.3, 3.4.4, 3.4.5, 3.5.2, 3.5 .3, 3.5.4, 3.5.5).

3. Also, in paragraph 2.5, you should indicate the sequence of primers used to amplify 16S DNA and 18S DNA.

These remarks do not diminish the importance of the work.

Sincerely yours, reviewer.

Author Response

  1. Perhaps it is necessary to calculate and present the indices of alpha diversity of microbial and fungal communities (Shannon, Chao1, etc.).

These data, as well as the results of the analysis, have been added to the corresponding sections of the paper: Lines 223–236 and lines 374–384.

  1. The methods used for data analysis should be more fully described: how analyzes were carried out for the analysis of principal components, correlations, molecular ecological network structure (sections 3.4.2, 3.4.3, 3.4.4, 3.4.5, 3.5.2, 3.5 .3, 3.5.4, 3.5.5).

Thank you for your suggestion. We have added a description of the correlation analysis method prior to sections 3.4.2–3.4.5 and highlighted it yellow; however, since 3.4 describes the same method, we have not repeated it.

  1. Also, in paragraph 2.5, you should indicate the sequence of primers used to amplify 16S DNA and 18S DNA

The sequences of generic primers have been added to this section.
